# Design, Synthetic Strategies, and Therapeutic Applications of Heterofunctional Glycodendrimers

**DOI:** 10.3390/molecules26092428

**Published:** 2021-04-22

**Authors:** Leila Mousavifar, René Roy

**Affiliations:** Glycosciences and Nanomaterial Laboratory, Université du Québec à Montréal, P.O. Box 8888, Succ. Centre-Ville, Montréal, QC H3C 3P8, Canada; leilyanmousavifar@gmail.com

**Keywords:** carbohydrates, dendrimers, glycodendrimers, vaccines, cancer, theranostics, immunodiagnostics, dynamic combinatorial chemistry, virus

## Abstract

Glycodendrimers have attracted considerable interest in the field of dendrimer sciences owing to their plethora of implications in biomedical applications. This is primarily due to the fact that cell surfaces expose a wide range of highly diversified glycan architectures varying by the nature of the sugars, their number, and their natural multiantennary structures. This particular situation has led to cancer cell metastasis, pathogen recognition and adhesion, and immune cell communications that are implicated in vaccine development. The diverse nature and complexity of multivalent carbohydrate–protein interactions have been the impetus toward the syntheses of glycodendrimers. Since their inception in 1993, chemical strategies toward glycodendrimers have constantly evolved into highly sophisticated methodologies. This review constitutes the first part of a series of papers dedicated to the design, synthesis, and biological applications of heterofunctional glycodendrimers. Herein, we highlight the most common synthetic approaches toward these complex molecular architectures and present modern applications in nanomolecular therapeutics and synthetic vaccines.

## 1. Introduction

Glycodendrimers constitute an important, albeit specific, family member in the field of dendrimers in general. Since their inception in the early 1990s, their chemistry has steadily evolved to address complex biological problems [1,2,3,4,5,6,7]. Through their precise and monodisperse architectural 3D structures, they have been used in most of the fields usually tackled with conventional dendrimers. However, because the glycan moieties of glycodendrimers, usually exposed as surface groups, belong to the family of natural biomolecules, they are directly implicated in a plethora of potential therapeutic aspects. This first and partial review has been systematically organized to present the various medical applications in which they are involved.

In their original design, glycodendrimers were established as a simpler synthetic version of complex multiantennary glycans found on cell surface glycoproteins. For instance, mannosylated dendrimers can mimic the oligomannopyranosides exposed on pathogens such as viruses, bacteria, and fungi [8,9]. Given that mammalian immune cells contain receptors for these mannosides (e.g., DC-SIGN) [10], they have been the target of viral attacks. A classic example has been observed in the case of HIV-I infections since the virus heavily exposed such complex saccharides within their gp120 glycoproteins [11]. As another example, the reversed multivalent carbohydrate–protein interactions have been exploited in the case of uropathogenic *E. coli* infections that rely on the bacterial FimH pili that bind to the mammalian self-oligomannopyranoside of urothelial glycoproteins as a cause of initial adhesion and colonization [12]. Consequently, glycodendrimers are valuable therapeutic nanomaterials to inhibit pathogen infections [13,14]. In addition, carbohydrate binding proteins are usually also multivalent (antibodies, lectins, galectins) or clustered on the cell surfaces. These complex and multiple interactions have been the subject of thorough investigations which have given rise to the “glycoside cluster effects” [15,16]. Glycodendrimers, by virtue of their cross-linking abilities (Figure 1) have been used in haemagglutination and quantitative precipitation assays.

As such, monodisperse and well-defined glycodendrimers represent ideal tools to study multivalent carbohydrate–protein interactions as they can be used to better understand the effects of carbohydrate density, flexibility, cross-linking ability, multivalency or steric hindrance due to proximity effect [17,18,19,20]. This initial account will focus on a relatively novel family of glycodendrimers that have been designed to further expose heterofunctionalities, either in the form of different surface carbohydrates (Janus glycodendrimers) or as glycodendrons possessing additional biological features (immunogenic peptides, probes, etc.). The synthetic strategies leading to these heterofunctional molecular architectures will be first examined using random dendrimer glycations and combinatorial approaches.

Figure 2 illustrates a wide variety of heterofunctional glycodendrimers studied thus far. For instance, preformed and commercially available dendrimers such as PAMAM can be randomly glycated to afford glycodendrimers of type (**1**) [21,22]. Two differently glycated dendrons can be linked to afford Janus-type glycodendrimers (**2**) [23]. Cyclodextrins (α, β, γ) have also been equally used as multivalent scaffolds (**3**) [24,25]. Additional cases of heterofunctional glycodendrimers have been disclosed using an “onion peel” approach that has provided precise alternating glycans (**4**) [26,27,28,29] instead of random distribution and numbers. Janus glycodendrimers possessing other aromatic scaffolds (**5**) have also been described [30]. Structure **6** with either probes or immunogenic peptides are useful as MRI contrast agents [31] or as vaccine candidates [32,33], respectively. Finally, cyclic peptides as scaffolds are interesting vaccine candidates (**7**) [34]. The next section will present applications in immunotherapy including vaccines, immunodiagnostics, and cell targeting.

## 2. Randomized Heterofunctional Glycodendrimers and Dynamic Combinatorial Library

Carbohydrate binding proteins of non-immune origin such as plant and bacterial lectins are naturally oligomeric [35] and are consequently excellent models for the study of multivalent carbohydrate/glycodendrimer–protein interactions [17,18,19,20]. Multivalent interactions are the key to several physiological events and their investigations have led to varied architectural design using a wide range of scaffolds [1,2,3,4,5,6,7,8,9]. Polyamidoamine (PAMAM) dendrimers, owing to their commercial availability, have been extensively used for this purpose. In this review’s context, heterofunctional PAMAM-glycodendrimers from generation three (G3) to six (G6) with 32, 64, 128, and 256 surface amine groups, respectively, have been modified with α-D-glucopyranosides (α-Glc), α-D-galactopyranosides (α-Gal), and α-D-mannopyranosides (α-Man) using the homotetrameric (at pH 7) plant lectin Concanavalin A from the jackbean *Canavalia ensiformis* (ConA) [21,22].

Hence, allyl glycosides 8–10 have been treated under radical initiated thiol–ene reaction catalyzed by 1,1-di(*t*-butylperoxy)cyclohexane in the presence of Boc-protected aminoethanethiol **11** to afford peracetylated sugar **12**, **14**, and **16** (Scheme 1). Boc-deprotection with TFA in DCM and treatment with thiophosgene led to isothiocyanates **13**, **15**, and **17**, respectively. Each sugar was then sequentially added in varied molar ratios in DMSO at room temperature to the PAMAM dendrimers from G3 to G6 and blocking the remaining free amine groups with PEG-dimer having an isocyanate function at one end. After sugar de-*O*-acetylation under standard Zemplén conditions (NaOMe, MeOH) and purification by dialysis and ultrafiltration, heterofunctional glyco-PAMAM-dendrimers were obtained and characterized by ^1^H-NMR and MALDI-TOF mass spectrometry. The necessity to add the PEG-dimer resulted from the lack of complete sugar modifications due to steric factors, even under forcing conditions (temperature, excess sugar isothiocyanates).

These randomly modified heterofunctional glycodendrimers were evaluated with the tetrameric ConA using relative potency values by hemagglutination inhibition assays using erythrocytes together with precipitation assays owing the cross-linking abilities of the glycodendrimers (see discussion above). It is well established that α-Man binds four times better than α-Glc, while α-Gal does not bind at all. The data allowed determining whether monovalent differences in affinity affect multivalent association constants in predictable ways. The results indeed suggested that multivalency can be influenced in predictable and in tunable ways. Clearly, monovalent differences were amplified by multivalent associations, and the mixtures of low- and high-affinity ligands could be used to attenuate multivalent binding activities. An analogous study (see below with β-cyclodextrin) further supports the synergistic effects of heteroclusters due to the secondary binding interactions with the “non-active” ligands. These combined studies confirmed a previous finding from this author that has been coined “subsite-assisted binding interactions” [36].

Dynamic combinatorial chemistry (DCC) has been introduced to rapidly and efficiently provide chemical libraries of bioactive compounds with limited efforts. The library components are generated through the rapid equilibrium of reversible reactants and products. This principle has been applied for the first time in carbohydrate chemistry by the Lehn’s group using disulfide interchanges [37] and aldehydes/hydrazides equilibrium to generate acylhydrazones [38]. This useful strategy has been successfully used to afford heterofunctional clusters (Scheme 2). In their preliminary attempts, the researchers generated a small set of 4-aminophenyl glycosides such as the β-D-galactoside **18** and the α-D-mannoside **19** which, upon amidation to varied dithiodicarboxylic acids (**20**, *n* = 2 or 3) using 1-ethyl-3-(3-dimethylamino)-propyl carbodiimide (EDC), afforded the initial carbohydrate dimers **21** and **22**. Mild disulfide interchanges catalyzed by dithiothreitol (DTT) at pH 7.4 were initiated under two sets of conditions coined “adaptive or pre-equilibrated libraries”. After the initiation, libraries were formed under both conditions of the scrambling process. The best performing clusters were selected using the homotetrameric plant lectin ConA either present during the library generation or simply added after the equilibrium was achieved, as discussed above for the Manno-PAMAM libraries (Scheme 1). As expected, due the preference of ConA toward mannopyranosides, the manno-dimers (**22**) were shown to be preferentially bound to the lectin. Interestingly, the scrambling interchanges in the presence of the lectin afforded slightly higher amounts of the manno-dimers, indicating that protein-templated dynamic combinatorial chemistry efficiently occurred.

An additional approach by the same group generated an analogous DCC process using a limited set of 4-formylphenyl glycosides **24**–**29** together with di- and tri-hydrazides **30**–**36** (Scheme 2B) for the formation of a library of acylhydrazones [38]. In this way, using again the plant lectin ConA as protein binder, they identified trimer **37** as the optimal ligand using relative inhibition in an enzyme-linked assay (ELLA) with yeast mannan as a coating ligand, a procedure developed in the author’s laboratory [39]. Trimer **37** had an IC_50_ of 22 μM, a 36-fold improvement when compared to methyl α-D-mannopyranoside (Me-Man) with an IC_50_ of 800 μM and a value comparable to the natural trimannoside (60-fold better than Me-Man).

## 3. Immune Cell Targeting, Immunodiagnostics, and Vaccines

### 3.1. Heterofunctional Glycodendrimers as Clearing Agents Following Radioimmunotherapy

Tumor-associated carbohydrate antigens (TACAs), originating from either glycolipids (gangliosides) [40] or *O*-linked mucin glycoproteins (MUCs) [41,42,43,44], have been extensively used as targeting agents for cancer immunotherapy. Therefore, and not surprisingly, the field of glycodendrimers, with their intrinsic multivalency and high affinity (avidity), has been exploited in the creation of powerful tools to provide therapeutic applications against cancer [44] that also include theranostics [45]. Several strategies can be applied to address this issue, amongst which, immune cell targeting through their well-studied mannoside receptors such as DC-SIGN [10], anti-carbohydrate antibodies and vaccines [40,41,42,43,44], screening microarrays using dendrimer’s increased sensitivity [46,47], and notably anti-cancer vaccines.

Interestingly, to this arsenal of glycodendrimers of therapeutic values against cancer, there is an additional avenue that was recently investigated. It has been referred to as clearing agents (CAs) that are used following radioimmunotherapy (RIT) to remove the excess of unwanted and noxious radioactive agents with the help of sugar dendrimers binding to liver receptors for catabolism [31]. RIT uses a monoclonal antibody (mAb) labeled with a radionuclide to deliver cytotoxic radiation to cancerous target cells [48]. A few examples consisted of yttrium (^90^Y)-ibritumomab tiuxetan (Zevalin), iodine (^131^I)-tositumomab (Bexxar), or lutetium (^177^Lu)-lilotomab satetraxetan (Betalutin) [49]. The radionuclide is usually bound to the mAb of interest through a chelating agent such as S-2-(4-aminobenzyl)-1,4,7,10-tetraazacyclododecane tetraacetic acid (DOTA, **40**) (Scheme 3).

Therapeutic indices (TIs; tumor-to-normal tissue-absorbed dose ratios) of radioimmunotherapy (RIT) should be maximized for the safe and effective treatment of solid tumors. Typically, however, RIT with radiolabeled-IgG antibodies suffers from low TIs due to the unfavorable pharmacokinetics of the IgG carrier, slow localization in the targeted tumor, rapid and high uptake in reticuloendothelial tissues. Consequently, the treatments are often ineffective at maximum tolerated activities. Therefore, glycodendritic CAs can remove nonlocalized targeting mAbs from circulation for hepatic catabolism, thereby enhancing the therapeutic index (TI).

Scheme 3 describes the synthesis of a fully synthetic glycodendrimer-based CA for DOTA-based pretargeted radioimmunotherapy (DOTA-PRIT). The novel glycodendron−CA (**41**) consists of a nonradioactive yttrium-DOTABn molecule (**40**) attached via a linker to a glycodendron displaying 16 terminal α-thio-N-acetylgalactosamine (α-SGalNAc) units. The methyl esters of dendron **38** were first hydrolyzed and then treated with peracetylated α-SGalNAc possessing an amine linker (**39**) using conventional amide coupling (HATU, DIPEA, DMF). The Boc-protected focal point of the resulting glycodendron was treated with TFA followed by coupling with the isothiocyanated DOTA precursor **40** to afford sugar-protected, thiourea-linked DOTA-glycodendron. Subsequent yttrium chelation with Yttrium(III) chloride hexahydrate under slightly acidic conditions and sugar deprotection under Zemplén conditions (NaOMe, MeOH) gave the desired CA agent **41**. The authors concluded that this novel heterofunctional glycodendron-DOTA CA complex (**41**) could be used for the enhanced blood clearance of an iodine-131 (^131^I) anti-cancer mAb together with having a high level of therapeutic index (TI).

The heterofunctional CA was developed for a chelate-based pretargeting strategy (DOTA-PRIT) for the theranostic imaging and treatment of solid tumors. The principle behind CAs is based on the use of bispecific-IgG antibodies having one binding portion to target the solid tumors together with an anti-DOTA function. The methodology of DOTA-PRIT consisted of separate, temporally spaced injections of three reagents: (1) a tetravalent bispecific IgG-single chain mAb with high affinity for (a) a tumor antigen (the IgG portion) and (b) a DOTA radiometal complex of yttrium; (2) the clearing agent (CA) to rapidly reduce circulating mAb after sufficient time is given for the antibody to accumulate at antigen-positive tumor; and (3) a radiolabeled DOTA-hapten such as the DOTA complex with the theranostic (i.e., γ- and β-emitting) isotope lutetium-177 ([^177^Lu]-DOTAn). To target the radioactivity to the tumor, the low-molecular weight circulating DOTA-hapten rapidly enters the tumor parenchyma and binds to the intratumoral mAb and is otherwise rapidly cleared via renal excretion.

### 3.2. Heterobifunctional Cancer Vaccines

As stated above, owing to their over expression in a number of cancer cells, TACAs from the group of *O*-linked mucins have been the subject of intense activity to generate anti-carbohydrate cancer vaccines [41,42,43,44]. The group of Bay et al. reported a fully synthetic anti-cancer vaccine for human use [50,51]. The new vaccine prototype, named MAG-Tn3 (**42**), is a tetrameric glycopeptide grafted on L-lysine dendrimer incorporating trimeric residues of the carbohydrate tumor-associated Tn antigen (α-D-GalNAc-O-R) and a CD4^+^ T-cell peptide epitope extracted from the highly immunogenic sequence of tetanus toxin (Figure 3). One major obstacle to vaccination is the high degree of MHC polymorphisms in the human population. Potential broad coverage can, however, be potentially accomplished by using promiscuous T-helper epitopes recognized by several of the common MHCs. One such “universal” T-helper cell epitope has been identified from tetanus toxin (TT). It is derived from the TT_830–844_ peptide sequence. Hence, the above group used such tetanus toxin-derived peptide segment as CD4^+^ T-cell epitope. The study successfully demonstrated that the vaccine-induced Tn-specific mouse antibodies mediated the killing of human Tn-positive tumor cells. The synthetic vaccine is currently being investigated in breast cancer patients (Phase I clinical trial). For the clinical trials, the vaccine was formulated with the recent GSK AS15 immuno-stimulant (CpG7909, monophosphoryl lipid A (MPL), and QS21) as a liposomal formulation. It has been shown to stimulate both innate and humoral responses. Patients with localized breast cancer with a high-risk of relapse were immunized with the above vaccine cocktail. The initial results of the clinical trial demonstrated that patients developed high levels of Tn-specific antibodies. These antibodies specifically recognized Tn-expressing human tumor cells which were killed through a complement-dependent cytotoxicity mechanism (CDC). This work illustrated that heterofunctional dendritic carbohydrate-bearing nanoparticles (NPs) could be attractive cancer vaccines. An analogous TACAs vaccine has been similarly described in the author’s laboratory [52]. It incorporated a related disaccharide known as the Thomsen–Friedenreich antigen (Galβ1-3GalNAc-α-*O*) and the work has been previously reviewed [33].

An appealing additional strategy was recently published that targeted both Tn and TF antigens [34]. Since most TACAs antitumor vaccines published thus far contained only a single target carbohydrate antigen resulting in the incomplete destruction of heterogeneously glycated cancer cells, the authors sought to incorporate two of the most studied TACAs described above. To this end, they used their typical orthogonal chemoselective ligation strategy to prepare fully synthetic glycosylated cyclic peptide scaffolds grafted with both Tn (**46**) and TF (**47**) antigens (Scheme 4). To this end, cyclic peptide **43** harboring four aldehyde groups, prepared by solid-phase peptide synthesis (SPPS), was coupled to orthogonally protected aminooxy derivative **44** to provide hexadecavalent scaffold **45** containing orthogonal functionalities (aldhehyde/azide) (Scheme 4). The alternatively substituted aldehydo-azide **45** was then treated with aminooxy α-D-GalNAc (Tn antigen) derivative **46** using TFA as catalyst. This was followed by a copper catalyzed cycloaddition (CuAAC) between the azide-substituted intermediate (not shown) and an alkynyl disaccharide (TF antigen) **47**, which afforded the hexadecavalent vaccine precursor **48**.

The ability of **48** to be recognized as a divalent tumor antigen was unambiguously demonstrated by direct ELISA assays using a known anti-Tn monoclonal antibody 9A7 [34]. Although the heterovalent structure showed binding capacities to 9A7 (mAb), the presence of the second TF epitope did not interfere with the recognition of Tn. Unfortunately, the study did not show the result of reversed anti-TF antibody data. This heterovalent glycosylated scaffold can therefore constitute an attractive strategy for the design of new generation vaccines once grafted to a suitable T helper cell epitope similar to those described previously through the Lys indicated by the arrow on **48**.

### 3.3. Blocking Antibody Formation to Prevent Autoimmune Diseases and Allergy

Cell surface glycans on pathogens or cancer cells can act as key antigens for the stimulation or development of innate as well as adaptive immunity. Synthetic glycoconjugates have been used in vaccines to mimic the presentation of these antigenic glycans which can elicit the production of specific antibodies. Carbohydrates are also crucial recognition features in the up- or downregulation of immune responses. Amongst immune cells, B cells are a necessary component of cellular immunity and autoimmune disease and rely on specific molecular signals for their regulation. The recognition and binding of specific glycoantigens to the B-cell receptors (BCR) results in B-cell activation, proliferation, and antibody production [33,43,44]. Recognition of α-(2→3) and α-(2→6)-linked sialic acids, commonly found on bacterial capsular polysaccharides and cancer cells by Siglec-G and CD22 co-receptors, are known to weaken or even abolish B-cell activation and are important for the development of immune tolerance. Therefore, this situation can block unwanted antibody formation and could thus be useful to prevent autoimmune diseases and allergy. It has been observed that the co-localization of BCR and CD22 leads to B-cell deactivation when the two receptors are clustered together with heterofunctionalized ligands. Glycoconjugates that can simultaneously display BCR antigens and CD22 ligands might induce B-cell tolerance by interacting with both receptors [53]. There have been several examples of multivalent homogeneous and bifunctional ligands designed to promote B-cell tolerance. These studies have raised the possibility of designing glycoconjugates as tolerance-inducing molecules, termed tolerogens.

A recent investigation pointed to the development of antibodies to the ABO human blood group antigens (HBGA) as tolerogens [53]. The elimination of anti-ABO antibodies in both organ and bone marrow transplantation would expand the safe use of blood products. The ABO system is characterized by the expression of ABH carbohydrate structures on human erythrocytes and other tissues. ABO incompatibility is a major challenge for blood transfusion and organ transplantation due to non-self anti-A or anti-B antibodies. The authors considered that a tolerogen displaying an A or B antigen together with a Siglec ligand could be effective for tolerance induction in the advance of transplantation. They described a conjugation method based on amine coupling alone or in combination with CuAAC and a heterotrifunctional linker. The method can yield well-defined multivalent heterofunctional scaffolds simultaneously containing synthetic carbohydrate antigens and Siglec ligands. The strategy enabled the authors to generate a panel of structurally well-defined multivalent homogeneous glycoconjugates of high molecular weight (12–18 kDa) (Scheme 5).

In Scheme 5, the authors reported an orthogonal ligation strategy in the preparation of complex oligosaccharides harboring four copies of one or two separate glycan antigens, providing 4–8 carbohydrate residues on a tetravalent poly(ethylene glycol) scaffold (Scheme 5). The approach afforded complex glycoconjugates approaching the size of glycoproteins (15–18 kDa) while remaining well defined. The synthetic strategy makes use of three orthogonal functional groups (**50**), including a reactive N-hydroxysuccinimide (NHS)-ester moiety on the linker to install the first carbohydrate epitope via reaction with a sugar amine such as **49** (blood group A type II). A fluorenylmethyloxycarbonyl (Fmoc)-protecting group was used as a masked amine functionality on the linker, which after standard deprotection (20% piperidine, DMF), allowed the NHS-activated tetrameric pentaerythritol (**52**) possessing a define poly(ethylene glycol) (PEG) spacer (60 residues, PDI 1.025). The azide group in the linker **51** was then used to incorporate the second carbohydrate epitope as the CD22 ligand (sialyl α-(2→6)-lactose) (**54**) via an alkyne-azide cycloaddition catalyzed by copper powder (CuAAC) to generate octameric-heterobifunctional glycoconjugate **55**.

To demonstrate the ability of heterobifunctional glycoconjugate **55** to interact simultaneously with both the BCR and CD22 cell surface receptors, the authors examined its ability to co-cluster the targeted receptors in vitro and modify their distribution. They chose an established cell line (A-BCL) previously used to characterize ABO-antigen coated nanoparticles that express a BCR complex that binds the blood group A antigen (type I and type II) but not the B antigens. Using confocal microscopy, they proved that the distribution of receptors was generally in isolated microclusters on untreated cells, while the BCR complex was found in larger but sparser clusters. After confirming that these cells expressed both receptors in isolated clusters, they showed that conjugate **55** modified the distribution of these receptors on the cell surface and is influential in their co-clustering. The treatment of cells with a control caused no detectable changes in CD22 or BCR organization. The work clearly supported the application of their strategy for investigating the cellular response to carbohydrate antigens of the ABO blood group system with the potential to prevent antibody formation, a useful situation in blood group transfusion.

### 3.4. Dendritic Glycopeptides as Vaccines against Allergy

Olive pollen is an important cause of respiratory allergy and peptide-based vaccines based on representative allergens represent promising therapeutic approaches. However, typical peptides require adjuvants to strengthen the weak immunogenicity of small peptides which can be overcome by using adjuvants, such as Toll-like receptor (TLR) ligands and glycoconjugates based on glucose and particularly mannose because of their cognate receptors (Dectin and DC-SIGN) on immune cells. These adjuvants modulate adaptive immune responses by cellular activation which have been progressively proposed for use in vaccine design. Heterobifunctional mannosylated glycodendropeptides (**66**) containing the 22 amino acids peptide sequence 109−130 from the most prevalent allergen of olive pollen (*Olea europaea*) (Ole e 1) (OE109–130) have been shown to represent a promising tool in the design of novel vaccines against olive pollen allergy due to their properties as promoters of the T lymphocyte regulatory cells (Treg) but not as activators of effector cells, efficient transport across the epithelial barrier, and no cytotoxicity (Scheme 6) [32].

Toward achieving these vaccine candidates, the “onion peel” type [26,27,28,29] dendron approach has been used (Scheme 6). Typically, tris-propargylated scaffold having a diethylene glycol linker **56** was treated with 2-azidoethyl α-D-mannopyranoside **57** under Copper(I)-catalyzed azyde-alkyne Cycloaddition (CuAAC) conditions (H_2_O/DMSO, CuSO_4_, NaAsc, TBTA) to afford glycodendron trimer **58**. After chloride to azide substitution (NaN_3_, DMF), intermediate **59** was again treated under similar CuAAC conditions, but using a triethylene glycol linker **60** to provide nonameric glycodendron precursor **61**. Analogous chloride to azide exchange, **62** was treated under slightly different CuAAC conditions (DMSO, CuBr, Tris((1-benzyl-4-triazolyl)methyl)amine (TBTA)) with linker **63** to give a maleimido-ending glycodendron **64**. Coupling with synthetic olive allergenic peptide OE109−130 possessing a cysteine residue at the N-terminal (**65**) under Michael thiol–ene afforded the desired vaccine candidate **66** with the suitable properties described above.

None of the glycodendrimers exhibited cytotoxicity in humanized cell lines. Confocal images indicated that mannosylated glycodendropeptides exhibited lower colocalization with a lysosomal marker. Moreover, mannosylated glycodendropeptides showed a higher transport tendency through the epithelial barrier formed by appropriate cell cultures. Finally, mannosylated glycodendropeptides **66** promoted Treg and IL10+Treg proliferation and IL-10 secretion by peripheral blood mononuclear cells (PBMCs) from allergic patients. Consequently, mannosylated dendrimers conjugated with OE109−130 peptide from Ole e 1 were identified as suitable candidates for the development of novel vaccines of olive pollen allergy.

### 3.5. Immunodiagnostics Using Glycan Microarrays

The accurate diagnostics and vaccine design for cancer and viral diseases such as AIDS, influenza, and SARS-COV-2 are of prime interest in glycosciences [54]. Therefore, the chemical characterization and syntheses of the precise carbohydrate epitopes to determine the often multivalent architectural presentation of these antigenic determinants have become a major goal. The glycotopology and deciphering the glycocode [55,56] may significantly influence the strategy of drug design. For instance, carbohydrate epitopes on viruses represent attractive targets for the development of carbohydrate-based vaccines [11,57,58,59]. Appreciating the exposure of carbohydrate epitopes on the cell surface allows to more reliably mimic the natural environment in the context of vaccine design. Of particular interest, the HIV envelope glycoprotein gp120 contains a plethora of high-mannose oligosaccharides [8,9] on its surface to shield peptides from recognition by the host immune system and facilitate invasion by binding to the C-type lectin DC-SIGN on dendritic cells. The incomplete mimicry of carbohydrate epitopes on the cell surface could lead to a failed vaccine design. Recently, Danishefsky and co-workers have shown that the Man_9_GlcNAc_2_-based vaccine elicited a high-titer antibody response that indeed recognized the Man_9_GlcNAc_2_ epitope but failed to neutralize HIV, thereby suggesting that it was not an optimal mimic of the epitope of gp120 [58].

To better understand the effects of carbohydrate density, flexibility, multivalency or steric hindrance due to proximity effect, Wong et al. [59] used heterofunctional dendritic glycan microarrays. To control the ratio of the mixed glycans more precisely, Wong and co-workers attached heterogeneous glycans to an AB_3_-type second generation dendrimeric scaffold (TRIS) (Figure 4). Two glycans, Man_4_ and Man_9_, were conjugated to the scaffold at different ratios to give a set of oligomannose constructs analogous to **67**. These oligomannose dendrons were then printed onto an NHS-activated glass slide to form an array of conjugates with various densities. Amongst these, heterogeneous oligomannose glycodendron **67** had the strongest binding affinity (K_D_ 13 nM) to 2G12 HIV-anti-gp120 antibody among the five oligomannose dendrons synthesized.

Interestingly, they found that heterogeneous glycans, prepared by exposing two distinct mannose oligosaccharides, and spotting onto glass slides, provided superior binding affinity compared to the individual components in the microarray experiments. In addition, the data suggested that heterogeneous ligand glycans could serve as a novel strategy for the development of precision carbohydrate-based vaccines. Their work should benefit the future design of carbohydrate-based vaccines by additional combinatorial approaches.

## 4. Conclusions

This review constitutes part one of a series dedicated to the design, synthesis and applications of heterobifunctional glycodendrimers. Obviously, because of its limited length, it has thus far covered only a small section of the literature on this important topic of nanomaterial and glycosciences. For this, we apologize to other key contributors whose work has not yet been covered. In addition to a few of the selected examples included into this account, there are several other applications equally important and they will be covered in subsequent reviews. Of particular interest are the contributions dealing with multivalent anti-adhesins against various pathogenic agents and microarrays. Several reviews have already covered glycodendrimers in general and a few have been cited in this work. However, we concentrated our efforts uniquely to heterofunctionalities, an area of activity that clearly deserves further development. We deeply acknowledge the scientific contributions, enthusiasm, and friendship of the dendrimer’s pioneers, and for this particular occasion, we wish to celebrate the 80th birthday anniversary of one among them, Dr. Jean-Pierre Majoral, to whom this Special Issue has been dedicated, guest edited by Prof. Ashok Kakkar and Dr. Anne-Marie Caminade, whose contributions to the field of dendrimers have also been inspiring.

## Data Availability

The data presented in this study are available on request from the corresponding author.

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
