# Peer review of "Design, Synthetic Strategies, and Therapeutic Applications of Heterofunctional Glycodendrimers"

_molecules, 2021, doi:10.3390/molecules26092428_

Round 1
Reviewer 1 Report
Design, Synthetic Strategies, and Therapeutic Applications of Heterofunctional Glycodendrimers
Authors: L. Mousavifar & R. Roy
Synopsis:
This review article focuses on the development of heterofunctional glycodendrimers for biomedical applications, and discusses the different scaffolds utilized as well as an array of strategies for synthesizing these complex glycopolymers.
In general, I found the review to be very interesting to read and I liked the examples presented. However, in general the manuscript suffered from organization flaws, both within the sections and in individual sentences. I would have given a score of moderate revision needed, had this been an option, as I do believe that the issues are significant enough that the authors must address them prior to the journal accepting the review. I have made some recommendations below that I believe will improve the flow of the document and improve the clarity for the readers. I have listed these under “major issues.” All other grammatical, typographical and organizational issues are listed under “minor issues.”
Major issues:
- Section 3 has an awkward flow to it. I think to improve the clarity of this section, the following changes should be considered. First, the discussion of the synthesis (paragraph 4) should be moved to follow paragraph 2. Second, the last sentence of paragraph 2 (lines 170-172) should be moved to the start of the 3rd The second part of the 4th paragraph (starting with the beginning of the sentence on line 195) should be a new paragraph and can be combined with the 3rd paragraph (move this part at the end), as this is where the biological activity of the CA is described. This would then provide an improved flow of information from general characteristics of TACAs/CAsàsynthesis CAsàbiological activity CAs and so on for other examples
- Lines 276-280: I would recommend adding a sentence here to give more insight into why it might be desirable to lower B cell activity and antibody production (something about autoimmune/allergy response).
- Lines 311-316: Not clearly expressed how compound 55 influenced CD22/BCR clustering. Looks like you mention the untreated cells and the control w/out 55 only. Please elaborate.
- Lines 317-346: Similar issue as what is described in the first bullet point in this section. I think that the authors need to re-work the organization of the section dealing with olive pollen to have the information flow in a more logical fashion. I would recommend: general properties of olive allergyàglycodendrimer properties/synthesis to generate vaccine against olive pollenàbiological data on glycodendrimer 66 and conclusions/future directions. I think this format would be easier to follow for the interested readers.
- Lines 357-386: Same issue. I would recommend re-organizing this section so it reads from generalàspecific in terms of content. I might also recommend that the part about Wong’s work in the first paragraph be moved to second paragraph to keep that specific example and corresponding information together instead of having part of it in one paragraph and part in another. I would also discuss the construction of the glycodendrimer before the biological assays/results, as this would make more sense and improve the flow of the section.
Minor issues:
- Line 16: Impetuous should be impetus
- Line 55: well-define should be well-defined
- Scheme 2-A: put DTT and pH 7.4 over the equilibrium arrows
- Line 155: insert comma after therefore
- Line 170: insert 40 within scheme 3 parentheses to indicate compound number for DOTA
- Line 198 delete antibody after mAb (redundant)
- Line 214/235: Scheme 4 should be labeled Figure 2
- Line 221 “currently investigated” should instead read “currently being investigated”
- Lines 240-243: sentence is awkward and unclear.
- Line 258: change “previously at the arrow pointing position onto 48” to “previously through the Lys indicated by the arrow on 48”
- Line 264: need a transition sentence or subheading to change topics.
- Line 268: change “are necessary” to “are a necessary”
- Line 273: add a comma after co-receptors
- Line 293: Begin paragraph, “In Scheme 6,” to provide a better transition
- Line 317: need a transition sentence or subheading to change topics.
- Lines 339: You need to define CuACC
- Line 340: define TBTA
- Line 357: need a transition sentence or subheading to change topic
- Line 362: change drugs to drug
- Lines 372-376: Unclear and awkward sentence. It might provide more clarity to separate this into two sentences.
- Line 381 and 389: Would be more appropriate as Figure 3
Author Response
This review article focuses on the development of heterofunctional glycodendrimers for biomedical applications, and discusses the different scaffolds utilized as well as an array of strategies for synthesizing these complex glycopolymers.
In general, I found the review to be very interesting to read and I liked the examples presented. However, in general the manuscript suffered from organization flaws, both within the sections and in individual sentences. I would have given a score of moderate revision needed, had this been an option, as I do believe that the issues are significant enough that the authors must address them prior to the journal accepting the review. I have made some recommendations below that I believe will improve the flow of the document and improve the clarity for the readers. I have listed these under “major issues.” All other grammatical, typographical and organizational issues are listed under “minor issues.”
Major issues:
- Section 3 has an awkward flow to it. I think to improve the clarity of this section, the following changes should be considered. First, the discussion of the synthesis (paragraph 4) should be moved to follow paragraph 2.
We modified accordingly and added several subheadings to section 3 to separate the various topics
- Second, the last sentence of paragraph 2 (lines 170-172) should be moved to the start of the 3rd
done
- The second part of the 4th paragraph (starting with the beginning of the sentence on line 195) should be a new paragraph and can be combined with the 3rd paragraph (move this part at the end), as this is where the biological activity of the CA is described. This would then provide an improved flow of information from general characteristics of TACAs/CAsàsynthesis CAsàbiological activity CAs and so on for other examples
This section was reorganized to follow the suggestion
- Lines 276-280: I would recommend adding a sentence here to give more insight into why it might be desirable to lower B cell activity and antibody production (something about autoimmune/allergy response).Therefore, this situation can block unwanted antibody formation and could thus be useful to prevent autoimmune diseases and allergy.
This was already clearly stated, but to further clarify it, we modified the sentence as follow:
"Therefore, this situation can block unwanted antibody formation and could thus be useful to prevent autoimmune diseases and allergy."
- Lines 311-316: Not clearly expressed how compound 55 influenced CD22/BCR clustering.
We added the following text where the two receptors were mentioned on new line 278-279:
……when the two receptors are clustered together with heterofunctionalized ligands….
- Looks like you mention the untreated cells and the control w/out 55 only. Please elaborate.
The text was abundantly modified for clarity as follow:
To demonstrate the ability of heterobifuntional glycoconjugate 55 to interact simultaneously with both BCR and CD22 cell surface receptors, the authors examined its ability to co-cluster the targeted receptors in vitro and modify their distribution. They chose an established cell line (A-BCL) previously used to characterize ABO-antigen coated nanoparticles that express a BCR complex that binds the blood group A antigen (type I and type II) but not the B antigens. Using confocal microscopy, they proved that the distribution of receptors was generally in isolated microclusters on untreated cells, while the BCR complex was found in larger but sparser clusters. After confirming that these cells expressed both, but isolated BCR and CD22 receptors, they showed that conjugate 55 modified the distribution of these receptors on the cell surface and is influential in their co-clustering. Treatment of cells with a control caused no detectable changes CD22 or BCR organization. The work clearly supported the application of their strategy for investigating cellular response to carbohydrate antigens of the ABO blood group system with the potential to prevent antibody formation, a situation useful in blood group transfusion.
- Lines 317-346: Similar issue as what is described in the first bullet point in this section. I think that the authors need to re-work the organization of the section dealing with olive pollen to have the information flow in a more logical fashion. I would recommend: general properties of olive allergyàglycodendrimer properties/synthesis to generate vaccine against olive pollenàbiological data on glycodendrimer 66 and conclusions/future directions. I think this format would be easier to follow for the interested readers.
Agreed: we moved the section “ None of the glycodendrimers exhibited…..” after scheme 6
- Lines 357-386: Same issue. I would recommend re-organizing this section so it reads from generalàspecific in terms of content. I might also recommend that the part about Wong’s work in the first paragraph be moved to second paragraph to keep that specific example and corresponding information together instead of having part of it in one paragraph and part in another. I would also discuss the construction of the glycodendrimer before the biological assays/results, as this would make more sense and improve the flow of the section.
Agreed again: we re-organized the sections – see in yellow
- Minor issues:
- Line 16: Impetuous should be impetus done
- Line 55: well-define should be well-defined done
- Scheme 2-A: put DTT and pH 7.4 over the equilibrium arrows done
- Line 155: insert comma after therefore done
- Line 170: insert 40 within scheme 3 parentheses to indicate compound number for DOTA done
- Line 198 delete antibody after mAb (redundant) done
- Line 214/235: Scheme 4 should be labeled Figure 2 done, we renumbered the remaining schemes
- Line 221 “currently investigated” should instead read “currently being investigated” done
- Lines 240-243: sentence is awkward and unclear.
We modified as:
Since most TACAs antitumor vaccines published thus far contained only a single target carbohydrate antigen resulting in incomplete destruction of heterogeneously glycated cancer cells, the authors tackled to incorporate two of the most studied TACAs described above.
- Line 258: change “previously at the arrow pointing position onto 48” to “previously through the Lys indicated by the arrow on 48” done
- Line 264: need a transition sentence or subheading to change topics.
We added a sub-section:
3.3 Blocking antibody formation to prevent autoimmune diseases and allergy
- Line 268: change “are necessary” to “are a necessary” done
- Line 273: add a comma after co-receptors done
- Line 293: Begin paragraph, “In Scheme 6,” to provide a better transition done
- Line 317: need a transition sentence or subheading to change topics.
Subheading added : 3.4 Dendritic glycopeptides as vaccines against allergy
- Lines 339: You need to define CuACC done
- Line 340: define TBTA done
- Line 357: need a transition sentence or subheading to change topic
Done with:
3.5 Immunodiagnostics using glycan microarrays
- Line 362: change drugs to drug done
- Lines 372-376: Unclear and awkward sentence. It might provide more clarity to separate this into two sentences.
Sentence modified as: To better understand the effects of carbohydrate density, flexibility, multivalency or steric hindrance due to proximity effect, Wong et al. [59] used heterofunctional dendritic glycan microarrays.
- Line 381 and 389: Would be more appropriate as Figure 3 done
Reviewer 2 Report
This review article (Part 1) on the design and synthetic strategic and therapeutic applications of heterofunctional glycodendrimers mimics is of utmost importance. The synthetic approaches to these target compounds are complex and low yields approaches. Authors have reviewed the alternative synthetic methodologies by randomized heterofunctional glycodendrimers and their dynamic combinatorial library from randomly distributed glucose, galactose and mannose residues as a viable route to these target molecules and it is of critical importance.
Additionally, synthesis of heterobifunctional cyclic glycopeptides using orthogonal ligation chemistry utilizing Tn and TF disaccharide moieties as antigens has been reviewed as well. This will definitely constitute the important goals and novelty of this review paper!
The article uses a specific chronology and classified the articles according the importance of topics.
The following suggested minor changes and recommendations should be introduced before the publication of the manuscript.
- Page 1, Abstract, line 15, replace “very” with “diverse”
- Page 7, line 187 and 204 ‘via” should be in italics.
- Page 11, scheme 6 should be placed on page 10 on the line 293.
- Page 13, line 386, ”the five oligomannose dendron synthesized” should either have numbers or literature references. Under current wording it is difficult to identify the oligomannose fragments and the work cited.
The review manuscript is of good quality and urgent importance and is well written and edited in order to meet the standard for the articles published in Molecules. Thus, I certainly recommend it for publication after the correction of these suggested minor changes.
Author Response
The following suggested minor changes and recommendations should be introduced before the publication of the manuscript.
- Page 1, Abstract, line 15, replace “very” with “diverse” done
- Page 7, line 187 and 204 ‘via” should be in italics. done
- Page 11, scheme 6 should be placed on page 10 on the line 293. done
- Page 13, line 386, ”the five oligomannose dendron synthesized” should either have numbers or literature references. Under current wording it is difficult to identify the oligomannose fragments and the work cited.
We modified the sentence in the following way:
Two glycans, Man4 and Man9, were conjugated to the scaffold at different ratios to give a set of oligomannose constructs analogous to 67. These oligomannose dendrons were then printed onto a NHS-activated glass slide to form an array of conjugates with various densities. Amongst these,......
Round 2
Reviewer 1 Report
Design, Synthetic Strategies, and Therapeutic Applications of Heterofunctional Glycodendrimers
Authors: L. Mousavifar & R. Roy
Synopsis:
This review article focuses on the development of heterofunctional glycodendrimers for biomedical applications, and discusses the different scaffolds utilized as well as an array of strategies for synthesizing these complex glycopolymers.
In general, I found the review to be very interesting to read and I liked the examples presented. The addition of the subheadings and reorganization of the paragraphs greatly improved the flow the review as well as the clarity. I only have a few minor edits to recommend below to complete prior to the manuscript’s publication.
Minor issues:
- Line 68: use “have” instead of “has”
- Line 71: delete “as scaffold” after the (5) as it is redundant
- Scheme 2: formula for ammonium formate is incorrect.
- Lines 218 & 239: should be figure 3, not figure 2
- Line 224: use “mediated” instead of “mediating”
- Line 251: change “harboring” to “containing” as harboring is redundant
- Line 307: Scheme 5 is redundant.
- Line 325: Instead of: “both, but isolated BCR and CD22 receptors” say “both receptors in isolated clusters,” for better clarity
- Lines 389 & 397: should be figure 4, not figure 3